# Increased perinatal survival and improved ventilation skills over a five-year period: An observational study

Ketil Størdal[1,2]*, Joar Eilevstjønn[3], Estomih Mduma[4], Kari Holte[1], Monica Thallinger[5], Jørgen Linde[6], Paschal Mdoe[4], Hussein Kidanto[7], Hege Langli Ersdal[6,8]

**1** Ostfold Hospital Trust, Grålum, Norway, **2** Norwegian Institute of Public Health, Oslo, Norway, **3** Laerdal Medical AS, Stavanger, Norway, **4** Haydom Lutheran Hospital, Manyara, Tanzania, **5** Vestre Viken HF, Gjettum, Norway, **6** Helse Stavanger HF, Stavanger, Norway, **7** Muhimbili National Hospital, Dar Es Salam, Tanzania, **8** SAFER, Stavanger, Norway

* ketil.stordal@fhi.no

**Data Availability Statement:** Data are available upon specific request. However, we are not allowed to make these openly available due to regulations from the National Institute of Medical Research in

## Abstract

### Background and aim

The Helping Babies Breathe program gave major reductions in perinatal mortality in Tanzania from 2009 to 2012. We aimed to study whether this effect was sustained, and whether resuscitation skills changed with continued frequent training.

### Methods

We analysed prospective data covering all births (n = 19,571) at Haydom Lutheran Hospital in Tanzania from July 2013 –June 2018. Resuscitation training was continued during this period. All deliveries were monitored by an observer recording the timing of events and resuscitation interventions. Heart rate was recorded by dry-electrode ECG and bag-mask-ventilation by sensors attached to the resuscitator device. We analyzed changes over time in outcomes, use of resuscitation interventions and performance of resuscitation using binary regression models with the log-link function to obtain adjusted relative risks.

### Results

With introduction of user fees for deliveries since 2014, the number of deliveries decreased by 30% from start to the end of the five-year period. An increase in low heart rate at birth and need for bag-mask-ventilation indicate a gradual selection of more vulnerable newborns delivered in the hospital over time.

Despite this selection, newborn deaths <24 hours did not change significantly and was maintained at an average of 8.8/1000 live births. The annual reductions in relative risk for perinatal death adjusted for vulnerability factors was 0.84 (95%CI 0.76–0.94).

During the five-year period, longer duration of bag-mask ventilation sequences without interruption was observed. Delivered tidal volumes were increased and mask leak was decreased during ventilation. The time to initiation or total duration of ventilation did not change significantly.

Tanzania. De-identified individual participant data will be made available to researchers whose methodologically sound proposal has been approved by the Scientific Steering Comitee for Safer Births Study Group. Proposals may be submitted to post@safer.net.

**Funding:** This work was supported by the Research Council of Norway and Laerdal Foundation for Acute Medicine to MT and JL. Laerdal Medicals provided support in the form of salary for Joar Eilevstjønn. The funders had no role in study design, data collection and analysis, decision to publish, or preparation of the manuscript. The specific roles of these authors are articulated in the 'author contributions' section.

**Competing interests:** Joar Eilevstjønn is a paid employee for Laerdal Medicals. There are no patents, products in development or marketed products associated with this research to declare. This does not alter our adherence to PLOS ONE policies on sharing data and materials.

## Conclusion

The reduction in 24-hour newborn mortality after introduction of Helping Babies Breathe was maintained, and a further decrease over the five-year period was evident when analyses were adjusted for vulnerability of the newborns. Perinatal survival and performance of ventilation were significantly improved.

## Introduction

The reduction of neonatal deaths over the last decade has been modest, despite declining mortality of infants between 1 month and five years of age [1]. The estimated global neonatal mortality ($\leq$ 28 days) fell from 31/1000 in 2000 to 18/1000 in 2017 [2]. The annual rate reduction in neonatal mortality was 3.1% compared to 4.7% after the neonatal period. Consequently, the relative contribution of neonatal deaths in the global under-5-mortality has increased from 39% to 45% [1]. The majority of these neonatal deaths are preventable. Safer births are achievable by improvements in a continuum from pregnancy, obstetric to neonatal care, each accounting for 1/3 of preventable deaths [3].

Intrapartum-related events, formerly referred to as birth asphyxia and defined as inadequate onset of spontaneous breathing after birth, are one of the three most prevalent causes of under-5-mortality [1]. The Helping Babies Breathe (HBB) program aimed to reduce death from intrapartum events in low- and middle-income settings, through training of health care workers in early identification and treatment [4].

In Tanzania, the roll-out of HBB from 2010 was associated with a 47% reduction in fresh stillbirths and early neonatal mortality over 24 months in eight large birth facilities [5]. Haydom Lutheran Hospital was one of the eight sites participating in the early roll-out of HBB. HBB training and research related to safer deliveries and newborn resuscitations continued in the Safer Births program [6]. Frequent low-dose training is one of the key elements to improve resuscitation skills [6, 7].

In Tanzania neonatal mortality decreased from 33/1000 in 2000 to 21/1000 in 2017 [2]. However, vital birth statistics are not available [8, 9], and estimates are inaccurate as almost half of the deliveries in Tanzania take place outside birth facilities. In a rural area with a low proportion of deliveries in hospital, this selection may preclude interpretation of temporal trends in mortality rates. It is therefore pertinent to adjust for vulnerability background factors in hospital deliveries.

After introduction of HBB in 2009–2012, efforts to sustain the initial improvements in perinatal mortality continued with onsite low-dose high-frequency HBB training. In the present study, we aimed to document changes in 24-hour neonatal mortality, fresh stillbirths and resuscitation performance during a five-year period from 2013 to 2018.

## Methods

Haydom Lutheran Hospital is situated in the rural area of the Manyara region of Northern Tanzania. The hospital has a catchment area of over 2 million inhabitants, and is the only hospital in the region providing comprehensive obstetric care with 250–400 deliveries per month. Haydom is a non-governmental hospital managed by the Lutheran church of Tanzania. All births in the hospital from 1st of July 2013 until 30th of June 2018 were recorded in the study.

This study was approved by the National Institute for Medical Research, Tanzania (NIMR/ HQ/R8a/Vol. IX/1434) and the Regional Committee for Medical and Health Research Ethics,

Western Norway (2013/110/REK). All health care providers gave informed oral consent. The mothers were informed but formal consent was not obtained for this observational study due to the urgency of mothers arriving in late stages of labor. This consent procedure was approved by the ethical committees.

## The initial HBB implementation

The local procedure for newborn resuscitation followed the HBB guidelines as implemented locally from 2009–2012 [5], emphasizing stimulation and early initiation of bag-mask ventilation (BMV) [4]. Newborn resuscitation was mainly the responsibility of midwives. Six labour rooms were each equipped with a resuscitation table. After caesarean section, newborn resuscitation was performed in a room adjacent to the operating theatre. Midwives determined if the newborn remained with the mother or was transferred to a neonatal ward after resuscitation. The neonatal ward provided basic care: respiratory support was limited to provision of supplemental oxygen with nasal prongs.

## The Safer Births program

The Safer Births program, starting in 2013, aimed to further facilitate HBB training and improve obstetric and newborn care by low-cost technology [6, 10]. During the study all midwives were encouraged to low-dose, high-frequency skills training as described by Mduma et al. [11]. Briefly, simulation training was performed on manikins, assisted by local-trainers who had undergone facilitator training for the HBB program. Visiting instructors from other places in Tanzania and Norway also participated regularly. The trainings were intended to take place weekly, but with some variations over time.

Trained data collectors observed and recorded the timing of events immediately after birth, and vital data and outcomes from all deliveries were recorded in a database.

Physiological data collected from 1st July 2013 until 30 Jun, 2018 included heart rate data recorded from ECG, and ventilation parameters obtained from sensors connected to the ventilation bag using a Newborn Resuscitation Monitor (Laerdal Global Health, Stavanger, Norway) (Fig 1). The arch-shaped ECG sensor was placed over the thorax or abdomen of the newborn. Two stainless-steel discs arranged on each side of the flexible arch acted as dry electrodes. The monitor used a hot-wire flow sensor connected to the ventilation bag similar to that in the Florian Respiratory Function Monitor (Acutronic Medical Systems, Hirzel, Switzerland). Volume was calculated by flow signal integration. Pressure was measured using an MPX5010 sensor (Freescale Semiconductor, Austin, TX, USA). The HR was displayed on the monitor installed in front of the resuscitation table visible to the provider, ventilation parameters were not displayed [12]. We analyzed data for a five years period with collection of physiological data until 30th of June 2018 (Fig 2, flow chart).

## Main and secondary outcomes

The main outcome was early neonatal mortality (ENM) defined as death before 24 hours of life among newborns delivered alive at Haydom Lutheran Hospital. As shown previously, live born babies with severe asphyxia may be misclassified as fresh stillbirths [13, 14]. Therefore, we also used the combined outcome of ENM and fresh stillbirths to analyze perinatal mortality.

The secondary outcomes were clinical performance defined by timing of events during resuscitation: Time from birth to start of BMV, the continuation of BMV by measuring duration of the first ventilation sequence before pause (>5 seconds interruption) and the fraction of time used for BMV once it had been initiated. The total duration of BMV was calculated as the time from the first to the last inflation. To assess actual bag-mask ventilation performance,

**Fig 1. Neonatal resuscitation monitor and its components used for data collection at Haydom Hospital during 1.7.13 to 30.6.18.**

we studied ventilation frequency, tidal volumes and mask leak the first five minutes of bag-mask ventilation.

## Main exposure and covariates

We analyzed by year (1[st] of July– 30[th] of June, starting 1[st] of July 2013) for the five-year period to assess effects of ongoing training in newborn resuscitation over time.

To assess for potential confounding, we obtained information from the prospective data-base of pregnancy complications, birth weight, prematurity, fetal heart rate abnormalities, the last recorded fetal heart rate and the need for BMV at birth and type of resuscitator used.

## Statistical methods

Descriptive analyses were performed by calculation of medians with interquartile ranges (IQR) or means with standard deviations according to their distribution. Categorical variables were analyzed using percentages and Chi-square test.

We used binary regression with log-link function to estimate relative risks for ENM by time [13, 15]. Risk factors which changed over time and were associated with the outcome (p<0.10) were included in a multivariate regression model if they changed the relative risk estimates by >10%. In the final model we therefore adjusted for birth weight, prematurity, fetal heart rate abnormalities, last recorded fetal heart rate and the need for BMV at birth. Cases with missing variables for any of the included adjustment variables (n = 15) were excluded from the adjusted model.

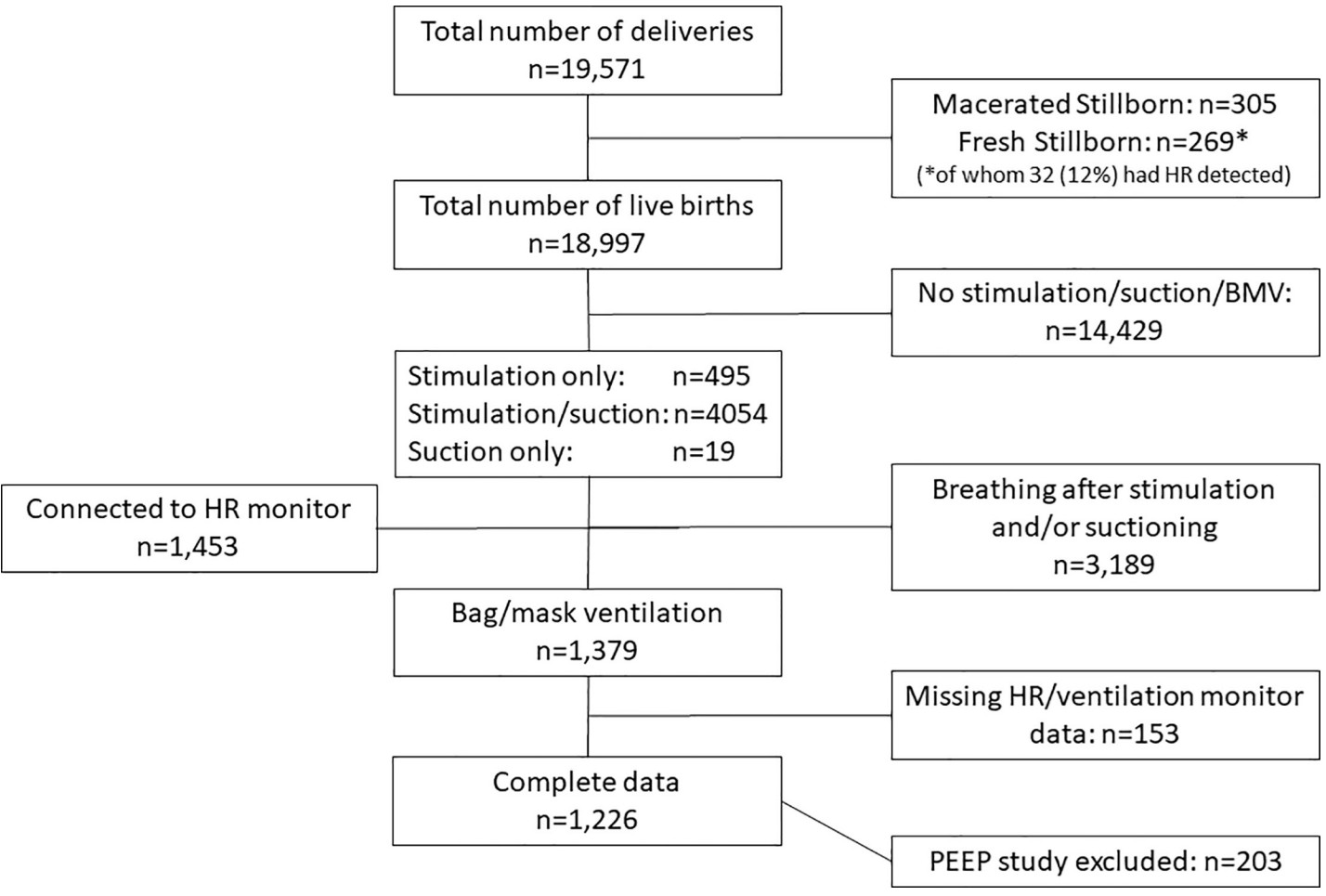

**Fig 2. Flow chart of participants in a population-based study of births in Haydom hospital during 1.7.2013 to 30.6.2018.**

We also performed sensitivity analyses excluding newborns who died without a history of resuscitation at birth because conditions other than intra-partum hypoxia may cause ENM.

Data were extracted using Matlab 2019a (MathWorks, Natick, MA) and analyzed using Stata SE 15 (StataCorp. 2017. Stata Statistical Software: Release 15. College Station, TX: StataCorp LP.)

## Results

During the five-year period, 18,997 infants were born alive, with a reduction over time from 4,662 to 3,197 annual births (Table 1). ENM fluctuated around 8.8/1000 live births, and the combined outcome of perinatal mortality (fresh stillbirths and ENM) around 22.6/1000 with no significant differences over time (Fig 3).

### Vulnerability factors over time

During the observation period, the proportion of deliveries characterized by risk factors for adverse outcome increased. The proportion of pregnancies with antenatal problems including preeclampsia increased significantly from 0.91% to 2.1% (p<0.001). The mean birth weight increased slightly, but an increasing proportion were <1500 g and preterm births increased from 2.5% to 3.5% (Table 1).

**Table 1. Baseline characteristics of live births at Haydom 1.7.2013–30.6.2018 (n = 18,997).**

| | | Period of birth | | | | |
|---|---|---|---|---|---|---|
| | 1st year | 2nd year | 3rd year | 4th year | 5th year | p-value* |
| | n = 4,662 | n = 3,909 | n = 3,565 | n = 3,664 | n = 3,197 | |
| **Maternal characteristics** | | | | | | |
| **No antenatal care, n (%)** | | | | | | 0.16 |
| | 44 (0.9) | 48 (1.2) | 36 (1.0) | 24 (0.7) | 32 (1.0) | |
| **Antenatal problem, n (%)** | | | | | | <0.001 |
| | 44 (0.9) | 29 (0.7) | 57 (1.6) | 65 (1.8) | 66 (2.1) | |
| **Preeclampsia/eclampsia, n (%)** | | | | | | 0.07 |
| | 21 (0.5) | 17 (0.4) | 25 (0.7) | 26 (0.7) | 28 (0.9) | |
| **Delivery** | | | | | | |
| **Mode of delivery, n (%)** | | | | | | <0.001 |
| Vaginal | 3,650 (78) | 2,961 (76) | 2,715 (76) | 2,770 (76) | 2,409 (75) | |
| Caesarean | 924 (20) | 943 (24) | 824 (23) | 858 (23) | 749 (23) | |
| Breech | 77 (2) | 0 (0) | 18 (0.5) | 19 (0.5) | 26 (0.8) | |
| Vacuum/others | 11 (0.2) | 4 (0.1) | 8 (0.2) | 17 (0.5) | 11 (0.3) | |
| **Fetal heart rate at admission, n (%)** | | | | | | <0.001 |
| Normal | 4,122 (88) | 3,510 (90) | 3,274 (92) | 3,444 (94) | 3,006 (94) | |
| Abnormal | 42 (1) | 33 (1) | 31 (1) | 38 (1) | 43 (1) | |
| Not detectable | 2 (0.04) | 0 (0) | 2 (0.1) | 5 (0.1) | 0 (0) | |
| Not measured | 496 (11) | 365 (9) | 257 (7) | 177 (5) | 145 (5) | |
| **Fetal heart rate during labour, n (%)** | | | | | | <0.001 |
| Normal | 3,917 (84) | 3,249 (83) | 2,973 (83) | 3,157 (86) | 2,751 (86) | |
| Abnormal | 143 (3) | 168 (4) | 168 (5) | 230 (6) | 298 (9) | |
| Not detectable | 0 (0) | 0 (0) | 7 (0.2) | 4 (0.1) | 1 (0.03) | |
| Not measured | 602 (13) | 494 (13) | 416 (12) | 272 (7) | 145 (5) | |
| **Last fetal heart rate before delivery (bpm), mean (SD)** | | | | | | 0.14 |
| | 134 (10) | 134 (10) | 134 (12) | 134 (14) | 133 (14) | |
| **Child characteristics** | | | | | | |
| **Birth weight, Mean (SD)** | 3215 (528) | 3253 (520) | 3280 (520) | 3377 (539) | 3342 (546) | <0.001 |
| <1500, n (%) | 24 (0.5) | 24 (0.5) | 15 (0.4) | 23 (0.6) | 28 (0.9) | |
| 1500–2500, n (%) | 325 (7) | 218 (6) | 199 (6) | 158 (4) | 136 (4) | |
| 2500–3499, n (%) | 2,895 (62) | 2,397 (61) | 2,071 (58) | 1,821 (50) | 1,690 (53) | |
| 3500–4499, n (%) | 1,370 (29) | 1,228 (31) | 1,242 (35) | 1,594 (44) | 1,284 (40) | |
| ≥4500, n (%) | 48 (1) | 42 (1) | 38 (1) | 68 (2) | 57 (2) | |
| **Gestational age < 37 weeks, n (%)** | | | | | | 0.042 |
| | 117 (2.5) | 103 (2.6) | 96 (2.7) | 121 (3.3) | 111 (3.5) | |
| **Female sex, n (%)** | | | | | | 0.049 |
| | 2,141 (46) | 1,882 (48) | 1,672 (47) | 1,692 (46) | 1,546 (48) | |

bpm: beats per minute

* p-values calculated with chi-square for categorical and binary regression for continuous variables

1st year: 1.7.2013–30.6.2014

2nd year: 1.7.2014–30.6.2015

3rd year: 1.7.2015–30.6.2016

4th year: 1.7.2016–30.6.2017

5th year: 1.7.2017–30.6.2018

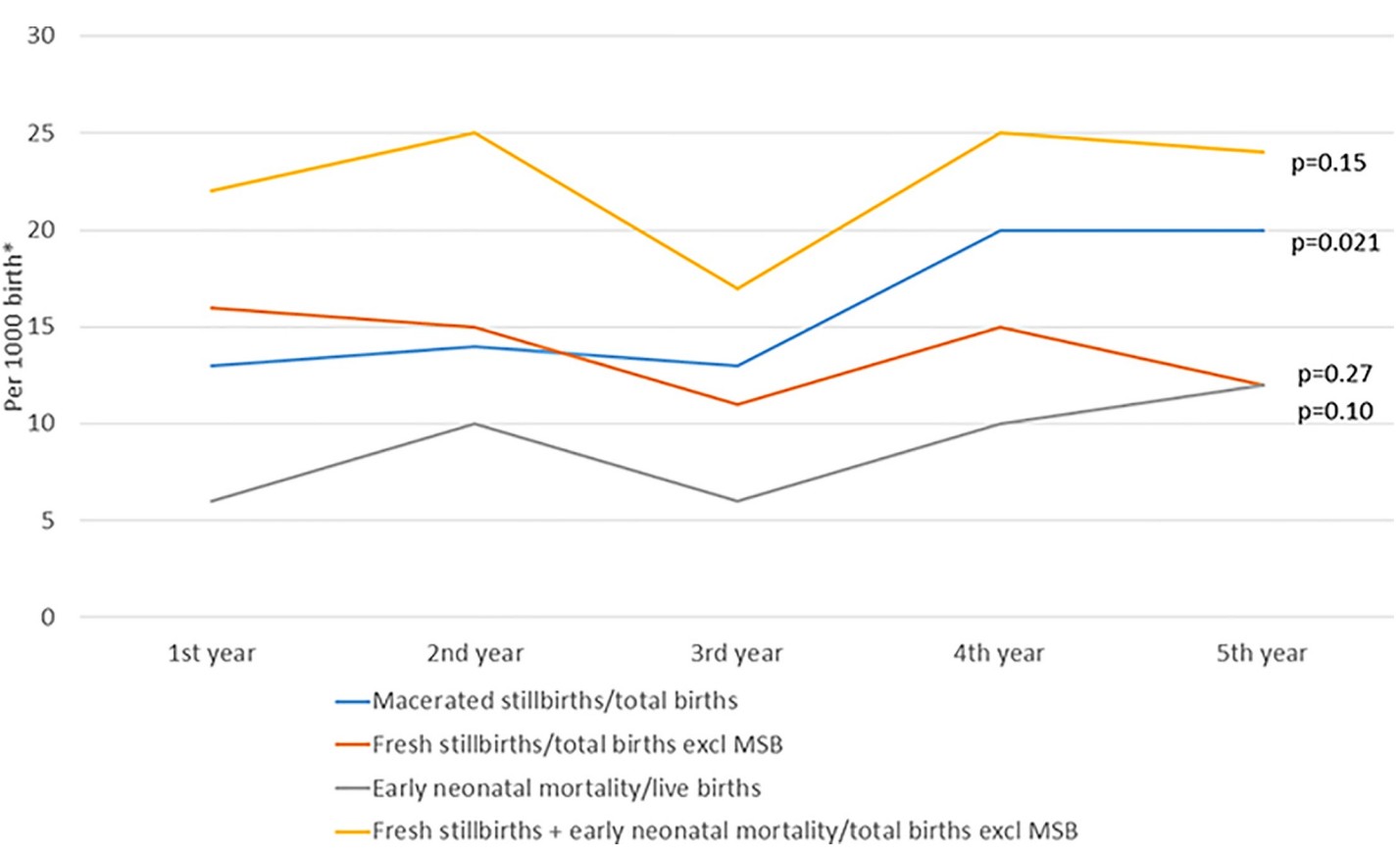

**Fig 3. Early neonatal mortality and stillbirths at Haydom Hospital by year during 1.7.13 to 30.6.18.**

Abnormal fetal heart rate during delivery increased gradually from 3% to 9%. The first detected heart rate among resuscitated babies fell from a median of 142 to 108 beats per minute (bpm), and the proportion with heart rate <100 bpm at birth increased from 8% to 22% (p<0.001, Table 2).

## Mortality adjusted for vulnerability

In unadjusted analyses, the ENM increased slightly during the five-year period (relative risk per year 1.11, 95%CI 1.00–1.24, Table 3). Vulnerability factors all increased during this period, as shown in Tables 1 and 2. After adjustment for abnormal fetal heart rate during delivery, birth weight, prematurity, initial heart rate and need for BMV at birth, there was no significant reduction in relative risk for ENM (relative risk per year 0.95, 95%CI 0.84–1.07, Table 3). Death the first 24 hours could be due to other reasons than intrapartum events, especially prematurity. Of those who died, 55/169 (33%) had not been ventilated, and thus would likely not be diagnosed as asphyxiated. We performed a sensitivity analysis without these 55: The unadjusted and adjusted analyses were essentially unchanged (Table 3).

The perinatal mortality showed no appreciable change during the five-year period in unadjusted analyses. However, after adjustment for vulnerability factors we found a significant annual decrease of 16% in perinatal mortality (relative risk per year 0.84, 95%CI 0.76–0.94, Table 3).

In the fully adjusted model, year of birth, fetal heart rate abnormalities, initial heart rate, birth weight, prematurity and need for BMV predicted perinatal death (Table 4).

**Table 2. Postnatal events and heart rate recordings of live births at Haydom 1.7.2013–30.6.2018 (n = 18,997).**

| | Period of birth | | | | | |
|---|---|---|---|---|---|---|
| | 1st year | 2nd year | 3rd year | 4th year | 5th year | p-value* |
| | n = 4,662 | n = 3,909 | n = 3,565 | n = 3,664 | n = 3,197 | |
| Apgar <5, n (%) | | | | | | |
| 1 min | 37 (0.8) | 50 (1.3) | 42 (1.2) | 45 (1.2) | 66 (2.1) | <0.001 |
| 5 min | 5 (0.1) | 7 (0.2) | 11 (0.3) | 20 (0.6) | 23 (0.7) | <0.001 |
| Stimulation/suction*, n (%) | | | | | | <0.001 |
| No stim./suction | 3,950 (85) | 3,234 (83) | 2,412 (68) | 2,684 (73) | 2,180 (68) | <0.001 |
| Stimulation only | 15 (0.3) | 36 (0.9) | 100 (2.8) | 159 (4.3) | 186 (6) | |
| Stim. and suction | 697 (15) | 641 (16) | 1053 (30) | 821 (22) | 843 (26) | |
| Bag/mask ventilation | 361 (7.7) | 240 (6.1) | 242 (6.8) | 272 (7.4) | 264 (8.8) | <0.001 |
| Heart rate | n = 389 | n = 298 | n = 274 | n = 262 | n = 232 | |
| Initial heart rate beats/min median (IQR) | 145 (97–179) | 142 (94–171) | 132 (84–174) | 118 (70–162) | 102 (67–148) | <0.001 |
| Initial heart rate in categories, n (%) | | | | | | |
| <60 | 32 (8) | 25 (8) | 21 (8) | 36 (14) | 50 (22) | <0.001 |
| 60–100 | 69 (18) | 58 (19) | 75 (27) | 74 (28) | 63 (27) | |
| 100–160 | 139 (36) | 98 (33) | 85 (31) | 80 (31) | 75 (32) | |
| >160 | 149 (38) | 117 (39) | 93 (34) | 72 (27) | 44 (19) | |
| Mean heart rate during 0–60 seconds after initial recording, median (IQR) | 143 (104–165) | 139 (101–165) | 139 (105–165) | 130 (91–157) | 129 (102–152) | <0.001 |

IQR: Inter-quartile range

* Chi2-test used for categorical analyses, linear regression for continuous analyses. P-values for trend.

## Changes in resuscitation performance

The proportion who received BMV varied from 6.1–7.7%, but increased to 8.8% the last year (Table 2). Use of stimulation increased substantially (Table 2). Time from birth to start of BMV was lowest the last year (median 99 seconds) but did not change significantly over time (p = 0.76, Table 5). The duration of the first BMV sequence increased over time from a median of 11 to 32 seconds (p<0.001, Table 5). Once initiated, BMV was continued for an increasing part of the first 60 seconds, from a median of 24 seconds the first compared to 48 seconds the last year (p<0.001, Table 5). However, the overall duration of ventilation from the first to the last inflation remained stable around a median of 130 seconds (range 127–142 seconds), explained by a higher number but shorter duration of BMV sequences in the first years (Table 5).

Median ventilation frequency decreased from >60 breaths/min the first year to 44–48 breaths/min the last three years of observation (p<0.001). The median weight-adjusted tidal volume, mean and peak airway pressures increased (p<0.001), whereas mask leak decreased slightly but significantly over time (p = 0.001, Table 5).

## Discussion

During a five-year period following a successful implementation of frequent HBB training, we found no further significant decline in ENM. A worrying decrease in hospital deliveries was observed, likely due to introduction of user fees since 2014 of 12 and 30 USD for vaginal and caesarean section deliveries, respectively. Consequently, hospitalized newborns included a higher proportion of infants with increased vulnerability for adverse outcomes. In analyses adjusted for vulnerability factors, perinatal deaths decreased significantly during the five years.

**Table 3. Relative risk for early neonatal death and perinatal death at Haydom 1.7.2013–30.6.2018.**

| | | Period of birth | | | | Overall linear | p-value[†] |
|---|---|---|---|---|---|---|---|
| Early neonatal deaths, n/total live births | 1st year 30/4,662 | 2nd year 41/3,909 | 3rd year 23/3,565 | 4th year 56/3,720 | 5th year 37/3,197 | | |
| **Unadjusted** | | | | | | | |
| | 1 (ref.) | 1.63 (1.02–2.61) | 1.00 (0.58–1.72) | 1.53 (0.94–2.47) | 1.63 (1.11–2.90) | 1.11 (1.00–1.24) | 0.04 |
| **Adjusted for abnormal fetal heart rate** | | | | | | | |
| | 1 (ref.) | 1.55 (0.97–2.48) | 0.93 (0.54–1.59) | 1.34 (0.83–2.18) | 1.44 (0.88–2.33) | 1.06 (0.95–1.18) | 0.31 |
| **Adjusted for birth weight and prematurity** | | | | | | | |
| | 1 (ref.) | 1.58 (1.00–2.48) | 1.00 (0.59–1.70) | 1.53 (0.95–2.44) | 1.72 (1.08–2.72) | 1.11 (1.00–1.23) | 0.05 |
| **Adjusted for initial heart rate** | | | | | | | |
| | 1 (ref.) | 1.72 (1.09–2.70) | 0.96 (0.56–1.63) | 1.34 (0.84–2.14) | 1.41 (0.88–2.26) | 1.04 (0.94–1.16) | 0.41 |
| **Adjusted for need for BMV** | | | | | | 1.09 | |
| | 1 (ref.) | 1.88 (1.19–2.98) | 1.10 (0.65–1.88) | 1.57 (0.98–2.52) | 1.72 (1.07–2.74) | 1.09 (0.99–1.21) | 0.39 |
| **Final adjusted model*** | | | | | | | |
| | 1 (ref.) | 1.88 (1.11–3.19) | 0.92 (0.50–1.69) | 1.25 (0.72–2.18) | 0.93 (0.53–1.64) | 0.95 (0.84–1.07) | 0.39 |
| **Sensitivity analysis excluding those never resuscitated (n = 55), adjusted** | | | | | | | |
| | 1 (ref.) | 1.95 (0.99–3.81) | 1.35 (0.66–2.73) | 1.39 (0.71–2.72) | 0.97 (0.49–1.91) | 0.96 (0.83–1.11) | 0.60 |
| **Perinatal deaths n/total births** | 106/4,738 | 99/3,967 | 62/3,604 | 92/3,720 | 77/3,235 | | |
| **Unadjusted** | | | | | | | |
| | 1 (ref.) | 1.12 (0.85–1.46) | 0.77 (0.56–1.05) | 1.11 (0.84–1.46) | 1.06 (0.80–1.42) | 1.01 (0.95–1.08) | 0.77 |
| **Final adjusted model*** | | | | | | | |
| | 1 (ref.) | 1.58 (1.01–2.47) | 0.88 (0.54–1.43) | 0.87 (0.55–1.39) | 0.53 (0.32–0.86) | 0.84 (0.76–0.94) | 0.002 |

* adjusted for antenatal problem, abnormal fetal heart rate during labour, birth weight, prematurity, initial heart rate and need for BMV after birth.

[†] p-value for trend

Excluded from adjusted analyses due to missing covariates: n = 15

Among several strengths in the study is the large size population-based study, performed in a rural setting of a low-income country with high burden of disease. The prospective design ensured high-quality data from the delivery and immediate post-natal period. Collection of physiological data during the first minutes of life provides information which is rarely accessible.

The selection of higher-risk deliveries during the period is evident, and accounted for as far as possible in the adjusted analysis. However, as in any observational study, residual confounding cannot be completely ruled out. A population-based birth registry with inclusion of all pregnancies including home deliveries would capture the complete picture and avoid selection bias. Specific causes of ENM were not possible to identify in our study, though in a sub-study from this cohort half of the deaths were related to intra-partum events [16]. We were, however, able to exclude deaths unlikely to be caused by perinatal hypoxia in a sensitivity analysis. An increased use of BMV ventilation over the 5-year period likely reflects the higher morbidity, and Apgar scores in this setting have previously been shown to have low prognostic value [17].

**Table 4. Predictors (relative risk with 95% confidence intervals) for perinatal death at Haydom Lutheran Hospital during five years (n = 19,264).**

| | Unadjusted | Adjusted* |
|---|---|---|
| Period, per year increase 2013–18 | | |
| | 1.01 (0.95–1.08) | 0.84 (0.76–0.94) |
| Fetal heart rate: | | |
| normal | 1 (ref.) | 1 (ref.) |
| abnormal | 5.7 (4.4–7.3) | 3.0 (2.0–4.5) |
| not detected | 74.0 (64.5–84.9) | 62.9 (20.5–193) |
| Birth weight | | |
| <1.5 kg | 28.7 (21.0–39.2) | 23.6 (10.5–53.2) |
| 1.5–2.5 kg | 7.7 (5.8–10.1) | 2.3 (1.3–4.1) |
| 2.5–3.5 kg | 1.4 (1.1–1.8) | 1.0 (0.7–1.6) |
| 3.5–4.5 kg | 1 (ref.) | 1 (ref.) |
| >4.5 kg | 3.1 (1.6–5.9) | 1.5 (0.5–5.1) |
| Prematurity | | |
| | 12.9 (10.7–15.6) | 5.7 (3.3–9.9) |
| Low initial heart rate after birth | | |
| <60 bpm | 165 (131–208) | 19.6 (11.1–34.9) |
| 60–100 bpm | 33.0 (23.3–46.6) | 1.8 (1.0–3.2) |
| 100–160 bpm | 6.3 (3.5–11.3) | 0.3 (0.2–0.8) |
| >160 bpm | 2.4 (0.99–6.0) | 0.2 (0.06–0.48) |
| No need for recording | 1 (ref.) | 1 (ref.) |
| Need for BMV after birth | | |
| | 74.0 (56.1–97.8) | 28.7 (16.0–51.2) |

*adjustments for fetal heart rate, birth weight category, prematurity, low initial heart rate and need for BMV after birth. In perinatal mortality results the initial heart rate and presumed need for BMV were set to the lowest category.

However, we cannot rule out that a sustained focus on HBB training may increase unnecessary resuscitation procedures.

During the nation-wide HBB roll-out during 2010/2011 in Tanzania, ENM in eight large hospitals including Haydom decreased from 13.4 to 7.1/1000. Fresh stillbirths decreased from 19 to 14.4/1000, and a remarkable fall of perinatal mortality of 34% was reported [5]. Before and after initial implementation of the HBB program ENM at Haydom was 11.1/1000 and 7.2/1000 respectively. The baseline perinatal mortality was 27/1000 and fell to 21.7/1000, and the decrease in ENM thus accounted for 74% of the reduced perinatal mortality [18]. In the present follow-up, our period of observation starts two years after regular HBB implementation and provides information on perinatal mortality, resuscitation performance and physiological data during a five-year period [6]. Our study shows that the initial reduction in perinatal mortality seen after HBB implementation has continued.

HBB has been introduced globally, and overall a reduction in perinatal mortality has been reported [19]. The variation between countries with highly successful implementation [5, 20, 21] to sites reporting no or modest effect on mortality [22, 23] is of note. The relative contribution of intra-partum hypoxia to ENM across countries may explain some differences, because a lower effect would be expected when other causes of ENM dominate. Besides, implementation and uptake of HBB could influence the results, and repeated low-dose onsite training is likely to facilitate successful implementation [11]. Several studies have reported improved resuscitation knowledge and practical skills after HBB training in a simulation setting [4, 24].

**Table 5. Bag-mask ventilation characteristics of live births at Haydom 1.7.2013–30.6.2018 (n = 18,997).**

| | Period of birth | | | | | |
|---|---|---|---|---|---|---|
| | 1ˢᵗ year | 2ⁿᵈ year | 3ʳᵈ year | 4ᵗʰ year | 5ᵗʰ year | p-value* |
| | n = 4,662 | n = 3,909 | n = 3,565 | n = 3,664 | n = 3,197 | |
| Bag/mask-ventilation, time events | | | | | | |
| | n = 358 | n = 234 | n = 240 | n = 269 | n = 262 | |
| Time from birth to BMV, seconds median,(IQR) | 105 (74–149) | 116 (81–164) | 119 (88–158) | 109 (74–162) | 99 (74–153) | 0.76 |
| Duration of first BMV sequence in seconds, median (IQR) | 11 (4–20) | 11 (5–22) | 16 (7–34) | 23 (9–52) | 32 (13–71) | <0.001 |
| Duration of BMV first 60 seconds in seconds, median (IQR) | 24 (15–36) | 28 (16–41) | 38 (22–51) | 46 (33–54) | 48 (35–59) | <0.001 |
| Duration of BMV sequences the first five minutes in seconds, median (IQR) | 12 (5–21) | 12 (5–24) | 16 (7–33) | 23 (10–45) | 27 (11–54) | <0.001 |
| Number of BMV sequences the first five minutes, mean (SD) | 4.2 (2.7) | 4.5 (2.7) | 4.1 (2.5) | 3.4 (2.2) | 3.1 (2.0) | <0.001 |
| Time from the first to the last inflation in seconds, median (IQR) | 132 (56–248) | 127 (59–340) | 127 (66–298) | 142 (64–282) | 130 (73–301) | 0.94 |
| Bag/mask ventilation characteristics† | | | | | | |
| | n = 293 | n = 247 | n = 238 | n = 173 | n = 114 | |
| Frequency of ventilations (inflations per minute) | 66 (50–84) | 52 (39–70) | 44 (35–53) | 48 (41–54) | 47 (41–53) | <0.001 |
| Tidal volume in ml/kg, median (IQR) | 5.3 (3.3–8.4) | 6.8 (4.3–10.0) | 7.5(5.0–11.5) | 6.6 (4.5–10.7) | 6.2 (3.2–9.8) | <0.001 |
| Mask leak in %, median (IQR) | 45 (33–62) | 46 (33–61) | 42 (30–57) | 41 (29–56) | 44 (30–57) | 0.001 |
| Mean airway pressure mbar, median (IQR) | 14 (12–17) | 13 (11–16) | 15 (12–18) | 17 (14–21) | 18 (16–21) | <0.001 |
| Maximum airway pressure mbar, median (IQR) | 34 (29–38) | 33 (28–36) | 36 (32–39) | 38 (34–43) | 40(37–45) | <0.001 |

IQR: Inter-quartile range.

BMV: Bag-mask ventilation.

* Chi2-test used for categorical analyses, linear regression for continuous analyses. P-values for trend.

† excluding participants in a randomized study using positive end-expiratory pressure in newborn resuscitation, n = 80 and n = 123 in 4ᵗʰ and 5ᵗʰ year, respectively. This study is reported and described elsewhere [34].

Two-step analysis: Mean of ventilation parameters for each newborn during first 5 minutes of active ventilation calculated, and in the second step this mean was used to calculate median/IQR.

Some loss of performance over time is observed [25–27], but skills retention was improved by frequent re-training and use of self-evaluation checklists [26]. Importantly, changes in clinical management do not necessarily follow simulated performance [28]. The present study documents improved resuscitation skills that are sustained over time, and is likely to improve perinatal mortality. Heart rate feedback during resuscitation may also have improved skills over time, as suggested in a qualitative study from Haydom [7].

In a study from Nepal, a reduction in the percentage of deaths attributable to intra-partum related complications from 51 to 33% was observed, but overall newborn mortality was unchanged [29]. This was also reported in the recent meta-analysis, pointing to the importance of a package of interventions needed to address the main causes of neonatal death [19]. In our study, one third of deaths occurred in newborns that did not receive BMV after birth. Intra-partum events accounted for about half the deaths before 7 days age at Haydom during 2014–17, but preterm birth and sepsis were among other preventable causes of death [16].

Our study is the first to report long-term changes in performance of resuscitation and BMV after HBB implementation. Analyses of actual ventilation performance demonstrate that practical skills improved over time with significantly longer duration of the first ventilation sequence and reduction in ventilation frequency targeting the recommended range. The increase in more complex patients may also change the resuscitation practice over time. The reduction in mask leak over time was desirable though admittedly small. Higher tidal volumes are associated with

more rapid HR increase in observational studies from Haydom [30, 31], but should be further studied in controlled trials before we can conclude whether this is recommendable.

In the initial HBB roll-out at Haydom, the use of suction/stimulation increased whereas use of BMV decreased significantly [11]. We found a further increase in suction/stimulation but also of BMV, which could be expected with increased vulnerability over time. In a meta-analysis of the effect of HBB training on resuscitation practices from four studies, the overall use of suctioning, stimulation or BMV was unchanged. However, interventions during the first Golden minute increased 2.5-fold [32]. In our study the increase during the five years of time used to ventilate the first minute after BMV was started is of note. The intensity and frequency of HBB training varied during the 5-year period, and due to shifts in the staff the quality of birth care and post-natal care is likely to be vulnerable. The substantial fluctuation over time in perinatal death rates, was however not reflected in the data on resuscitation performance.

In the present study the quality of immediate newborn care at birth was improved over time, probably due to continuous focus on frequent HBB training. However, coverage and access are likely even more important for the total perinatal mortality than quality of care in the hospital. These factors are interrelated, as increased quality in a delivery unit is likely to encourage more pregnant women to seek care. However, financial constraints may counteract this effect. In Tanzania in 2011/12, 44% had less than 1.25 USD per day for their daily living, and poverty was even more prevalent in rural districts [33]. In this context, 12 to 30 USD in hospital fees for delivery is a considerable burden to the family economy. Free delivery care should be a high priority, as provided in governmental hospitals in Tanzania.

In this study we have shown that neonatal resuscitation skills improved over time at a site with high focus on continued and frequent HBB training. The perinatal mortality adjusted for vulnerability factors decreased, whereas ENM showed a non-significant decline. Further reduction in neonatal mortality in Tanzania will likely require a comprehensive approach to increase facility births and address the major causes of death.

## Author Contributions

**Conceptualization:** Ketil Størdal, Hege Langli Ersdal.

**Data curation:** Joar Eilevstjønn, Monica Thallinger.

**Formal analysis:** Ketil Størdal, Joar Eilevstjønn.

**Funding acquisition:** Ketil Størdal, Hussein Kidanto, Hege Langli Ersdal.

**Investigation:** Ketil Størdal, Joar Eilevstjønn, Estomih Mduma, Kari Holte, Monica Thallinger, Jørgen Linde, Paschal Mdoe, Hege Langli Ersdal.

**Methodology:** Ketil Størdal, Joar Eilevstjønn, Hege Langli Ersdal.

**Project administration:** Joar Eilevstjønn, Estomih Mduma, Hussein Kidanto, Hege Langli Ersdal.

**Resources:** Hussein Kidanto, Hege Langli Ersdal.

**Software:** Joar Eilevstjønn.

**Supervision:** Estomih Mduma, Hege Langli Ersdal.

**Writing – original draft:** Ketil Størdal.

**Writing – review & editing:** Joar Eilevstjønn, Estomih Mduma, Kari Holte, Monica Thallinger, Jørgen Linde, Paschal Mdoe, Hussein Kidanto, Hege Langli Ersdal.

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
