## [Decision Letter · Decision Letter 0]

8 May 2020

PONE-D-19-30351

Increased perinatal survival and improved ventilation skills over a five-year period

PLOS ONE

Dear Dr Størdal,

Thank you for submitting your manuscript to PLOS ONE. After careful consideration, we feel that it has merit but does not fully meet PLOS ONE’s publication criteria as it currently stands. Therefore, we invite you to submit a revised version of the manuscript that addresses the points raised during the review process.

Please pay close attention to the reviewers comments and address their concerns, in particular those around overlap with previous mortality data. In addition, *PLOS ONE* does not copyedit accepted manuscripts, so the language in submitted articles must be clear, correct, and unambiguous. We recommend that authors seek independent editorial help before submitting a revision. These services can be found on the web using search terms like “scientific editing service” or “manuscript editing service.”

We would appreciate receiving your revised manuscript by Jun 22 2020 11:59PM. To enhance the reproducibility of your results, we recommend that if applicable you deposit your laboratory protocols in protocols.io, where a protocol can be assigned its own identifier (DOI) such that it can be cited independently in the future. For instructions see: http://journals.plos.org/plosone/s/submission-guidelines#loc-laboratory-protocols

We look forward to receiving your revised manuscript.

Kind regards,

Kelli K Ryckman

Academic Editor

PLOS ONE

Journal Requirements:

2. Please amend your current ethics statement to address the following concerns: Please explain why was written consent was not obtained, how you recorded/documented participant consent, and if the ethics committees/IRBs approved this consent procedure.

3. Thank you for stating the following financial disclosure: "No"

4. Thank you for stating the following in your Competing Interests section:  "No"

Reviewers' comments:

Reviewer's Responses to Questions

**Comments to the Author**

1. Is the manuscript technically sound, and do the data support the conclusions?

Reviewer #1: Yes

Reviewer #2: Yes

Reviewer #3: Partly

2. Has the statistical analysis been performed appropriately and rigorously? 

Reviewer #1: Yes

Reviewer #2: Yes

Reviewer #3: I Don't Know

3. Have the authors made all data underlying the findings in their manuscript fully available?

Reviewer #1: Yes

Reviewer #2: Yes

Reviewer #3: No

4. Is the manuscript presented in an intelligible fashion and written in standard English?

Reviewer #1: Yes

Reviewer #2: Yes

Reviewer #3: No

5. Review Comments to the Author

Reviewer #1: Thank you for the opportunity to review this paper entitled, “Increased perinatal survival and improved ventilation skills over a five-year period”.

This prospective analysis evaluates whether the effect of the HBB program was sustained over time.

They showed that the decrease 24h newborn mortality was sustained even improved after adjusting for covariables. They also noted an improvement on ventilation performance. The is a wealth of data collected in this project and showing feasible ways of improving neonatal survival is an important goal.

My major concern is that the is significant overlap in the mortality data with a previous published paper with only 1.5 years of additional data. I think that focusing more on the ventilation portion versus mortality would add more to the literature given that the mortality data on the majority of the mortality data of this data-set has already been published depicting the fact that the effect of the program can be sustained with frequent training.

Abstract: Predictors of survival can be added to the aims as this was a big focus. In the methods, some information regarding training schedule and data analysis can be added to the methods. I think the first 2 sentences in the results regarding the change in population can be condensed. I would remove the last sentence of the conclusion.

Introduction: I think adding the already known data regarding frequent training and the effect of HBB from reference 6 would be helpful in the intro.

Methods:

1. Though the frequency of the skills training is referenced to another paper, I think this paper would benefit from a brief description of the “low dose high frequency skills training” that is a major intervention thought to be responsible for the sustained outcomes. I think the same can be said for the monitoring set up – it is frustrating for me as the reader to have to reference several articles to determine the major determinants of the methods – though these can be referenced some details are required or the paper does not stand alone.

2. I would like to know from reading this paper alone the basics of the training and the set-up of the ventilation data collection. In addition I feel that the description of neonatal resuscitation performance was sparse and needed more detail.

3. I think that the adjustment for vulnerability factors should go under the statistical analysis section.

4. I recommend that the statistical section be reviewed by the statistician involved in the project as this section feels incomplete. For example, the categorical variable comparisons in the table was noted to be chi squared but this is not mentioned in the methods. In addition, your mortality risk ratios I would assume are based off of multi variable logistic regression given that mortality is a binary variable. Please also include the statistical software used.

Results:

1. The first sentence of the second paragraph read awkwardly.

2. In the second sentence please give exact numbers.

3. Did you do/can you add analysis to determine if there was a significant differences between specific time points versus just an overall difference as presented? This may be helpful.

4. I think graphic depiction of the mortality data versus the many tables would add to the paper.

5. It would be beneficial to break up table 3 as there is so much information. For the linear regression, was the outcome modeled per year? If so the r-square value would be useful in determining the strength of the association. As noted above, the methods in general need more clarification.

6. Was there a qualitative portion of the respiratory data – ie was BMV indicated when administered and appropriately stopped, were TV given appropriate for the size of the infant, etc

Discussion:

1. Generally the for me it makes sense for strengths and limitations to come right before the conclusion.

2. I don’t think the subheadings are needed

3. I do not think the 3rd paragraph under comparison with other studies adds much to the discussion, and could be removed.

4. Can you comment on the fact that continued exposure to higher acuity resuscitations may alone have some impact on these skills.

5. Can you say that this is the first study that shows the long-term sustained effects if your reference 6 by the same groups shows this as well (acknowledged that yours is an additional 18mo follow-up).

6. I think it would be helpful in the discussion to add more on the simulation/training aspect with the ventilation outcome with respect to other studies.

Reviewer #2: Nicely done paper. I've made comments on the attached file. In brief I had a few questions:

- Can you describe who provided the "low-dose, high-frequency skills training"? Was this done by local personnel, or did it involve visiting instructors from Europe, the USA, etc.?

- Can you provide data on emergency vs scheduled/repeat C-Section rates over time? This would be helpful in understanding the changing risk profile of your OB population.

- Can you comment on why year 2 was so much worse than other years (Table 4)?

- Do you know if the increased use of BVM was all due to appropriate and indicated use? For example, it appears that incidence of BVM was 4-fold the incidence of 1-min Apgar < 5, & 10-fold the incidence of 5-min Apgar < 5 (Table 3).

- Can you comment on any potential downsides of HBB, for instance, the incidence of Pneumothoraces?

Reviewer #3: Stordal and colleagues present an observational study reporting on perinatal survival and ventilation skills over a five year period of ongoing training following an initial intervention associated with the Helping Babies Breathe program.

Strengths of this study include the prospective nature, the large and relatively complete data obtained, and the inclusion of physiological data to assist in explaining the mortality outcomes. Key limitations of the study include confounding associated with a change in health funding that altered the complexity of neonatal admissions. This confounding required results to undergo statistical adjustment for increasing vulnerability of the infant case mix, to demonstrate an annual reduction in relative risk for perinatal death.

An annotated manuscript is uploaded to assist authors with their revisions. This manuscript makes some suggestions for improved text – but I would strongly recommend the authors to seek professional English editing: overall I think the text length could be reduced by at least 10 % with more efficient word use, and clearer sentence structure.

MAJOR COMMENTS:

The authors should consult the STROBE guidelines for reporting an observational study.

It would be preferable to make deidentified data available for transparency. De-identified data should not compromise any data privacy restrictions.

Specific statements regarding Financial Disclosure, Competing Interests and Ethics statements should be entered in line with suggested response types in the online submission form rather than just stating “no” or “as stated in the manuscript”.

TITLE: the study design should be indicated in the title and abstract

ABSTRACT

Background/aim: - the years of the initial program should be studied.

Methods should summarise statistical methodology including method used for adjustment of outcomes. rather than refering loosely to analysis of changes over time… (e.g., prospective observational cross-sectional cohort study).

Results – please see suggestions for adjusted wording of the last few sentences.

Conclusions – please see suggestions for adjusted wording. The final conclusion is not justified without data relating to population level neonatal mortality/morbidity over the same interval (i.e., was not being born in a hospital associated with increased neonatal mortality.

INTRODUCTION

Overall – the rationale for the study is not well presented. If the HBB decreased mortality, the authors need to justify why a follow-up study was necessary and what it aimed to achieve.

P9, Sentence 1 and 2: It would be helpful to present the fall in global mortality in both 1-5 years and neonatal population to support the statement in the first sentence that reduction in neonatal deaths has been slower over the last 2 decades

P9 final paragraph. This paragraph is unclear. The abstract suggests 1 follow-up study 2013-2018 and previous paragraphs in the introduction suggest the HBB program commenced in 2010 with the initial study extending for 24 months This paragraph suggests a study 2011-2016, and another study with ongoing onsite-low-dose HF HBB. The crossover in dates is confusing to the reader and it is not clear if this is a hybrid of the initial study or a separate follow-up study. I suggest the authors revise the wording of this paragraph to make it explicity clear when the five year study was completed and the timing of the initial study. This paragraph should provide a hypothesis for the study.

METHODS

Overall, the methodology section appeared to include elements of the previous study (2010-2011) and the current 2013-2016 study which was confusing to read. As this paper is about the 2013-2018 study, the methodology should exclusively relate to what was done in this study, with a citation used to reference back to the previous study.

Key elements of the study design and ethical approvals should be provided early in the methodology section.

The setting should simply state the hospital/location and relevant dates (including day not just month and year) for recruitment and data collection. Detail about fees for delivery and the effect on hospital delivery numbers should be part of a discussion.

Participants should be described including eligibility and exclusion criteria

Outcomes should be described separately to statistical methodology/treatment, and any explanation/justification of the methodology should go into the discussion.

Statistical elements that are missing include management of missing data, methods to examine subgroups/interactions, how confounding variables were managed, the approach to regression modelling and treatment of collinear variables (e.g., birthweight and gestation).

RESULTS

The exclusion of premature infants from the analysis is not well justified – death in the first 24 hours from prematurity is most often related to respiratory distress and may stem from insufficient resuscitation.

Some results are presented without accompanying statistics

It is not clear why infants involved in the PEEP trial were excluded from the ventilation analysis but not from the mortality outcomes, given PEEP may alter effectiveness of resuscitation and hence the resulting mortality.

Ventilation frequency should be presented as inflations/min or breaths/min not as 60/min

Tidal volume was corrected for weight and this should be clear in the results.

DISCUSSION

Paragraph 1: suggest that a full stop is provided after “user fees” and that the rest of the sentence becomes a new sentence as “Consequently, hospitalised newborns included a higher proportion of infants with increased vulnerability for adverse outcomes”.

Reference to the previous study would be better phrased as “before and after initial implementation of the HBB program was 11.1/1000 and 7.2/1000 respectively. The following sentence could be revised to indicate that reduction in early neonatal mortality accounted for the majority of the overall decrease in baseline perinatal mortality from 27/1000 to 21.7/1000 over the same period.

It is not clear why the authors say that the period of observation started 2 years after regular HBB implementation and extended for a further 1.5 years, when the actual study went for 5 years. The numbers don’t add up?

When other studies are quoted, the discussion would have more impact if the information was synthesised and presented as a comparison to the current data to indicate what the literature is telling us overall about the issue at hand. Reference to the study from Nepal does not indicate what the intervention was that led to a reduction of deaths from intra-partum complications but didn’t change overall neonatal mortality, and hence the reader is left confused about the relevance of this information.

It was not clear that the increase in (weight corrected) tidal volumes was due to the HBB training or due to the change in patient mix. It is odd that the increase in tidal volume during resuscitation was not matched by an increase in heart rate. How do the authors explain this?

MINOR COMMENTS

Numbers < 10 should be written in full

Overall, there are too many unnecessary abbreviations in the manuscript that make it difficult to read and digest.

Table 1 – data are usually presented as n (%) rather than % (n). As it is not clear what the numbers are for the child characteristics weight ranges it would be best to indicate in the table legend at the bottom of the table that data are presented as % (n) unless indicated otherwise. Any abbreviations (e.g., bpm) need to be explained in the table legend (which goes for other tables also – e.g., MSB in Table 2).

Table 5 needs to indicate what the numbers are – presume relative risk and 95 % CI?

6. PLOS authors have the option to publish the peer review history of their article (what does this mean?). If published, this will include your full peer review and any attached files.

Reviewer #1: No

Reviewer #2: No

Reviewer #3: No

---

## [Author Response · Author response to Decision Letter 0]

13 Aug 2020

We would like to thank the editors and reviewers for their work and valuable comments. The editors’ remarks have been accommodated. In the resubmitted version, we have incorporated changes according to the comments below, and in the following respond point-by-point to the reviewer’s comments. 

Reviewer #1: Thank you for the opportunity to review this paper entitled, “Increased perinatal survival and improved ventilation skills over a five-year period”.

This prospective analysis evaluates whether the effect of the HBB program was sustained over time.

They showed that the decrease 24h newborn mortality was sustained even improved after adjusting for covariables. They also noted an improvement on ventilation performance. The is a wealth of data collected in this project and showing feasible ways of improving neonatal survival is an important goal.

My major concern is that the is significant overlap in the mortality data with a previous published paper with only 1.5 years of additional data. I think that focusing more on the ventilation portion versus mortality would add more to the literature given that the mortality data on the majority of the mortality data of this data-set has already been published depicting the fact that the effect of the program can be sustained with frequent training.

The current manuscript adds to the previous paper by ventilation data over time, which have not been published before. In the revised manuscript. We have aimed to further highlight the changes in resuscitation performance stating clearly that this is one of the aims (introduction).

Abstract: Predictors of survival can be added to the aims as this was a big focus. In the methods, some information regarding training schedule and data analysis can be added to the methods. I think the first 2 sentences in the results regarding the change in population can be condensed. I would remove the last sentence of the conclusion.

We have included more information in the aim and the methods, and condensed the results to stay within word limits. As the study has several aims, we intend to narrow down and not include predictors of survival in the aim (end of introduction). We have removed the last sentence of the conclusion. 

Introduction: I think adding the already known data regarding frequent training and the effect of HBB from reference 6 would be helpful in the intro.

We have added one sentence and an additional reference (third paragraph, page 2):

“Frequent low-dose training is one of the key elements to improve resuscitation skills.6,7»

Methods:

1. Though the frequency of the skills training is referenced to another paper, I think this paper would benefit from a brief description of the “low dose high frequency skills training” that is a major intervention thought to be responsible for the sustained outcomes. I think the same can be said for the monitoring set up – it is frustrating for me as the reader to have to reference several articles to determine the major determinants of the methods – though these can be referenced some details are required or the paper does not stand alone.

Thank you for this suggestion. In the revised manuscript we have included more information regarding the training and monitor set-up as suggested. A figure of the monitor has been added (Fig 2). 

(page 4): Briefly, weekly simulation training was performed on manikins, assisted by local-trainers who had undergone facilitator training for the HBB program. Visiting instructors from other places in Tanzania and Norway also participated regularly.

(page 5): Physiological data collected from 1st July 2013 until 30 Jun, 2018 included heart rate data recorded from ECG, and ventilation parameters obtained from sensors connected to the ventilation bag using a Newborn Resuscitation Monitor (Laerdal Global Health, Stavanger, Norway) (Fig 1). The arch-shaped ECG sensor was placed over the thorax or abdomen of the newborn. Two stainless-steel discs arranged on each side of the flexible arch acted as dry electrodes. The monitor used a hot-wire flow sensor connected to the ventilation bag similar to that in the Florian Respiratory Function Monitor (Acutronic Medical Systems, Hirzel, Switzerland). Volume was calculated by flow signal integration. Pressure was measured using an MPX5010 sensor (Freescale Semiconductor, Austin, TX, USA). The HR was displayed on the monitor installed in front of the resuscitation table visible to the provider, ventilation parameters were not displayed. 

2. I would like to know from reading this paper alone the basics of the training and the set-up of the ventilation data collection. In addition I feel that the description of neonatal resuscitation performance was sparse and needed more detail.

Please refer to above corrections for training and monitor setup. Resuscitation performance has been detailed more (page 6, second paragraph).

“The secondary outcomes were clinical performance defined by timing of events during resuscitation: Time from birth to start of BMV, the continuation of BMV by measuring duration of the first ventilation sequence before pause (>5 seconds interruption) and the fraction of time used for BMV once it had been initiated. To assess actual bag-mask ventilation performance, we studied ventilation frequency, tidal volumes and mask leak the first five minutes of bag-mask ventilation.” 

3. I think that the adjustment for vulnerability factors should go under the statistical analysis section.

We have moved the description of adjustment variables including “vulnerability factors” to the statistics section as suggested. 

4. I recommend that the statistical section be reviewed by the statistician involved in the project as this section feels incomplete. For example, the categorical variable comparisons in the table was noted to be chi squared but this is not mentioned in the methods. In addition, your mortality risk ratios I would assume are based off of multi variable logistic regression given that mortality is a binary variable. Please also include the statistical software used.

The method for analysis of categorical date has been added (Page 6, last paragraph).

The regression method is not a logistic regression model with logit-link (which provides odds ratios) but as the study is a cohort design we have chosen to use a binary regression model with log-link (which provides relative risks). This information has been added to the text. Description of selection of adjustment variables has been moved and expanded in the revised manuscript, please refer to the manuscript for details. 

We have also included the statistical software: 

“Data were extracted using Matlab 2019a (MathWorks, Natick, MA) and analyzed using Stata SE 15 (StataCorp. 2017. Stata Statistical Software: Release 15. College Station, TX: StataCorp LP.)»

Results:

1. The first sentence of the second paragraph read awkwardly.

We have simplified the sentence to clarify: “During the observation period, the proportion of deliveries characterized by risk factors for adverse outcome increased.”

2. In the second sentence please give exact numbers.

This has been corrected. 

3. Did you do/can you add analysis to determine if there was a significant differences between specific time points versus just an overall difference as presented? This may be helpful.

We believe that adding more details would risk to overload the presentation. However, we kindly refer to the paper by Mduma et al (ref. 6) for more details in this respect. 

4. I think graphic depiction of the mortality data versus the many tables would add to the paper.

Thank you for this suggestion. In the revised manuscript the mortality data from Table 2 are replaced by a line diagram (Figure 3). 

5. It would be beneficial to break up table 3 as there is so much information. For the linear regression, was the outcome modeled per year? If so the r-square value would be useful in determining the strength of the association. As noted above, the methods in general need more clarification.

We agree that Table 3 (now Table 2 and 3) was large and that breaking up to display bag/mask-ventilation in a separate table provides better readability. For the new Table 2, only two of the analyses are linear, so we chose not to include the R square. 

6. Was there a qualitative portion of the respiratory data – ie was BMV indicated when administered and appropriately stopped, were TV given appropriate for the size of the infant, etc

This information would certainly be of interest. However, other studies from the research group have analyzed these data (video analysis of appropriate BMV: Haug et al, Clinical Simulation in Nursing, for Tidal volumes: J.E. Linde et al. Resuscitation 117 (2017) 80–86). 

Discussion:

1. Generally the for me it makes sense for strengths and limitations to come right before the conclusion.

This could be done but preferences differ among author, reviewers and editors. We feel the logical flow in the discussion is maintained without this change. 

2. I don’t think the subheadings are needed

We have removed the subheadings as suggested. 

3. I do not think the 3rd paragraph under comparison with other studies adds much to the discussion, and could be removed.

We have removed this sentence as suggested. 

4. Can you comment on the fact that continued exposure to higher acuity resuscitations may alone have some impact on these skills.

Thank you for this comment: In a qualitative study from Haydom hospital, the midwives have indicated that feedback from the monitor improves their skills. We have included a new sentence: 

“Heart rate feedback during resuscitation may also have improved skills over time, as suggested in a qualitative study from Haydom.7»

5. Can you say that this is the first study that shows the long-term sustained effects if your reference 6 by the same groups shows this as well (acknowledged that yours is an additional 18mo follow-up).

We have modified this sentence, as the current study adds more data and time but the previous study also has long-term data. Instead, we include that the study is the first to monitor changes in ventilation performance with physiological data over time.

“…and provides information on perinatal mortality, resuscitation performance and physiological data during a five-year period” 

6. I think it would be helpful in the discussion to add more on the simulation/training aspect with the ventilation outcome with respect to other studies.

We have added new text to several of the paragraphs to highlight the improved resuscitation performance over time, likely due to a high focus on this at the site. 

Reviewer #2: Nicely done paper. I've made comments on the attached file. In brief I had a few questions:

- Can you describe who provided the "low-dose, high-frequency skills training"? Was this done by local personnel, or did it involve visiting instructors from Europe, the USA, etc.?

We have included more information regarding the low-dose, high-frequency skills training in the revised version: Local personnel were trained as HBB-trainers, and some were also facilitated by visiting instructors.

“Briefly, simulation training was performed on manikins, assisted by local-trainers who had undergone facilitator training for the HBB program. Visiting instructors from other places in Tanzania and Norway also participated regularly. The trainings were intended to take place weekly, but with some variations over time.»

- Can you provide data on emergency vs scheduled/repeat C-Section rates over time? This would be helpful in understanding the changing risk profile of your OB population.

We have some information regarding planned vs emergency causes of C-Section over time. However, almost all C-Sections are emergency C-Sections, and the data regarding causes of emergency C-Sections do not provide sufficient quality to answer this question. 

- Can you comment on why year 2 was so much worse than other years (Table 4)?

We have also seen this fluctuations over time, with year 2 as an outlier. The resuscitation performance (Table 2 and 3) was relatively similar in year 2. The HBB training varied during the 5-year period, and with shifts in the staff the quality of birth care and post-natal care is likely to have varied though not directly reflected in the data on resuscitation performance. We have added comments on this in the discussion, underlying factors not recorded in our data are likely needed to explain this change.

“The intensity and frequency of HBB training varied during the 5-year period, and due to shifts in the staff the quality of birth care and post-natal care is likely to be vulnerable. The substantial fluctuation over time in perinatal death rates, was however not reflected in the data on resuscitation performance.”

- Do you know if the increased use of BVM was all due to appropriate and indicated use? For example, it appears that incidence of BVM was 4-fold the incidence of 1-min Apgar < 5, & 10-fold the incidence of 5-min Apgar < 5 (Table 3).

In this setting, research from our group has documented that the use of Apgar is highly inaccurate (Ersdal et al, Pediatrics 2012; 129: e1238-1243): 50% of those who died secondary to asphyxia had received a 5-minute Apgar score of ≥7. Thus, the Apgar scores are highly inaccurate and likely to explain this discrepancy. We have included a comment and reference to this finding. 

“An increased use of BMV ventilation over the 5-year period reflects the higher morbidity, and Apgar scores in this setting have previously been shown to have low prognostic value.”

- Can you comment on any potential downsides of HBB, for instance, the incidence of Pneumothoraces?

We unfortunately do not have access to portable chest x-rays at the hospital, so pneumothoraces are not possible to confirm accurately. Potentially, the focus on stimulation and suctioning from the start of the HBB program may have resulted in unnecessary suctioning that is now discouraged in resuscitation algorithms. We have added the following: 

“However, we cannot rule out that a sustained focus on HBB training may increase unnecessary resuscitation procedures.”

Reviewer #3: Stordal and colleagues present an observational study reporting on perinatal survival and ventilation skills over a five year period of ongoing training following an initial intervention associated with the Helping Babies Breathe program.

Strengths of this study include the prospective nature, the large and relatively complete data obtained, and the inclusion of physiological data to assist in explaining the mortality outcomes. Key limitations of the study include confounding associated with a change in health funding that altered the complexity of neonatal admissions. This confounding required results to undergo statistical adjustment for increasing vulnerability of the infant case mix, to demonstrate an annual reduction in relative risk for perinatal death.

An annotated manuscript is uploaded to assist authors with their revisions. This manuscript makes some suggestions for improved text – but I would strongly recommend the authors to seek professional English editing: overall I think the text length could be reduced by at least 10 % with more efficient word use, and clearer sentence structure.

Thank you for this comment, we have revised the text according to the annotated manuscript and aimed to make the revision more to-the-point. We have also revised the language according to the comments. 

MAJOR COMMENTS:

The authors should consult the STROBE guidelines for reporting an observational study.

It would be preferable to make deidentified data available for transparency. De-identified data should not compromise any data privacy restrictions.

Data are available upon specific request. However, we are not allowed to make these openly available due to regulations from the National Institute of Medical Research in Tanzania. 

De-identified individual participant data will be made available to researchers whose methodologically sound proposal has been approved by the Scientific Steering Comitee for Safer Births Study Group. Proposals may be submitted up to 36 months following article publication to hege.ersdal@safer.net.

Specific statements regarding Financial Disclosure, Competing Interests and Ethics statements should be entered in line with suggested response types in the online submission form rather than just stating “no” or “as stated in the manuscript”.

This has been corrected. 

TITLE: the study design should be indicated in the title and abstract

In the title we have now included: An observational study

ABSTRACT

Background/aim: - the years of the initial program should be studied.

The years of the initial program have been included. 

Methods should summarise statistical methodology including method used for adjustment of outcomes. rather than refering loosely to analysis of changes over time… (e.g., prospective observational cross-sectional cohort study).

In the revision we have added “…We analyzed changes over time in outcomes, use of resuscitation interventions and performance of resuscitation using regression models to obtain adjusted relative risks.

Results – please see suggestions for adjusted wording of the last few sentences.

In the revision we have changed the wording of the last few sentences to: “During the five-year period, longer duration of bag-mask ventilation sequences without interruption was observed. Delivered tidal volumes were increased and mask leak was decreased during ventilation.”

Conclusions – please see suggestions for adjusted wording. The final conclusion is not justified without data relating to population level neonatal mortality/morbidity over the same interval (i.e., was not being born in a hospital associated with increased neonatal mortality.

We have modified the wording of the conclusion, and removed the last sentence as we do not have population-based data of perinatal mortality for the region. 

“The reduction in 24-hour newborn mortality after introduction of Helping Babies Breathe was maintained, and a further decrease over the five-year period was evident when analyses were adjusted for vulnerability of the newborns.”

INTRODUCTION

Overall – the rationale for the study is not well presented. If the HBB decreased mortality, the authors need to justify why a follow-up study was necessary and what it aimed to achieve.

P9, Sentence 1 and 2: It would be helpful to present the fall in global mortality in both 1-5 years and neonatal population to support the statement in the first sentence that reduction in neonatal deaths has been slower over the last 2 decades

The phrasing has been changed according to suggestions. We have added accurate numbers for comparison between neonatal and post-neonatal mortality rates, please refer to the revised manuscript. 

P9 final paragraph. This paragraph is unclear. The abstract suggests 1 follow-up study 2013-2018 and previous paragraphs in the introduction suggest the HBB program commenced in 2010 with the initial study extending for 24 months This paragraph suggests a study 2011-2016, and another study with ongoing onsite-low-dose HF HBB. The crossover in dates is confusing to the reader and it is not clear if this is a hybrid of the initial study or a separate follow-up study. I suggest the authors revise the wording of this paragraph to make it explicity clear when the five year study was completed and the timing of the initial study. This paragraph should provide a hypothesis for the study.

We have changed the wording to clarify the periods of time included in our study: 

“After introduction of HBB in 2009-2012, efforts to sustain the initial improvements in perinatal mortality continued with onsite low-dose high-frequency HBB training. In the present study, we aimed to document changes in 24-hour neonatal mortality, fresh stillbirths and resuscitation performance during a five-year period from 2013 to 2018.”

METHODS

Overall, the methodology section appeared to include elements of the previous study (2010-2011) and the current 2013-2016 study which was confusing to read. As this paper is about the 2013-2018 study, the methodology should exclusively relate to what was done in this study, with a citation used to reference back to the previous study.

Key elements of the study design and ethical approvals should be provided early in the methodology section.

The setting should simply state the hospital/location and relevant dates (including day not just month and year) for recruitment and data collection. Detail about fees for delivery and the effect on hospital delivery numbers should be part of a discussion.

We have moved the section of ethical approvals to the end of the first paragraph. 

Further, we have divided with inclusion of separate headlines the text describing the initial HBB study and the Safer Births study with exact time period and interventions. The text has been revised according to suggestions from the reviewer, please refer to the “track changes” document. The reduction in births and user fees is moved to the discussion. 

Participants should be described including eligibility and exclusion criteria

All births in the hospital including stillbirths are recorded in the dataset. This is stated in the revised manuscript: “All births in the hospital from 1st of July 2013 until 30th of June 2018 were recorded in the study.” 

Outcomes should be described separately to statistical methodology/treatment, and any explanation/justification of the methodology should go into the discussion.

Statistical elements that are missing include management of missing data, methods to examine subgroups/interactions, how confounding variables were managed, the approach to regression modelling and treatment of collinear variables (e.g., birthweight and gestation).

The outcomes are described under a separate headline, now also including the secondary outcomes of resuscitation performance.

Exposure follows this section, as shown in the revised manuscript (“track changes”). We have added adjustment for type of bag used during the study period as explained. 

In the statistics section we have added information regarding handling of missing data, adjustment for confounding and regression models. Subgroup or stratified analyses could be done, but would add further complexity to the presentation. Therefore we prefer to adjust for subgroup characteristics (for example term/preterm birth). The term “gestational age” appears once in the results section, but the correct term should be “prematurity”: Gestational week was not used (this has not been recorded with sufficient precision), only as a dichotomous variable for preterm/term birth. With a Spearman rho of 0.2 between prematurity and birth weight, collinearity does not preclude the analyses. 

RESULTS

The exclusion of premature infants from the analysis is not well justified – death in the first 24 hours from prematurity is most often related to respiratory distress and may stem from insufficient resuscitation.

We include all – also preterms - in the main analyses. The analysis excluding those that had not been ventilated at all is performed as a sensitivity analysis only, and this is clarified in the revised manuscript. Whether any of these should have been ventilated is possible, however other causes of death including sepsis or severe prematurity are certainly likely in this category. We have modified the wording to clarify, and added the results from this sensitivity analysis to Table 3. 

Some results are presented without accompanying statistics

The numbers from statistical analyses have been added to the text where appropriate, in addition to the original version displaying this in Tables. 

It is not clear why infants involved in the PEEP trial were excluded from the ventilation analysis but not from the mortality outcomes, given PEEP may alter effectiveness of resuscitation and hence the resulting mortality.

Thank you for this important comment. The type of bag used did not predict mortality (p=0.25) and therefore did not satisfy the criteria for inclusion in the adjusted model. This is now explained in the Methods/statistics section.

Ventilation frequency should be presented as inflations/min or breaths/min not as 60/min

Tidal volume was corrected for weight and this should be clear in the results.

We have changed this in the revised manuscript. 

DISCUSSION

Paragraph 1: suggest that a full stop is provided after “user fees” and that the rest of the sentence becomes a new sentence as “Consequently, hospitalised newborns included a higher proportion of infants with increased vulnerability for adverse outcomes”.

We have revised this paragraph according to suggestions. 

Reference to the previous study would be better phrased as “before and after initial implementation of the HBB program was 11.1/1000 and 7.2/1000 respectively. The following sentence could be revised to indicate that reduction in early neonatal mortality accounted for the majority of the overall decrease in baseline perinatal mortality from 27/1000 to 21.7/1000 over the same period.

Thank you for this suggestion, we have rephrased and added the number showing that decrease in ENM accounted for most of the reduction in perinatal mortality: 

“The baseline perinatal mortality was 27/1000 and fell to 21.7/1000, and the decrease in ENM thus accounted for 74% of the reduced perinatal mortality”

It is not clear why the authors say that the period of observation started 2 years after regular HBB implementation and extended for a further 1.5 years, when the actual study went for 5 years. The numbers don’t add up?

We have clarified in the revised manuscript, stating time periods clearly in the Methods section and highlighting what is new in the present study. 

“In the present follow-up, our period of observation starts two years after regular HBB implementation and provides information on perinatal mortality and resuscitation performance during a five-year period” 

When other studies are quoted, the discussion would have more impact if the information was synthesised and presented as a comparison to the current data to indicate what the literature is telling us overall about the issue at hand. Reference to the study from Nepal does not indicate what the intervention was that led to a reduction of deaths from intra-partum complications but didn’t change overall neonatal mortality, and hence the reader is left confused about the relevance of this information.

We have added to the discussion points that we believe are relevant to explain why HBB implementation have different impact across countries (page 12 and 13). 

It was not clear that the increase in (weight corrected) tidal volumes was due to the HBB training or due to the change in patient mix. It is odd that the increase in tidal volume during resuscitation was not matched by an increase in heart rate. How do the authors explain this?

The data shown for HR in Table 2 are absolute numbers of the first recorded heart rate and during the first 60 seconds thereafter in order to characterize vulnerability of the newborns over time. Thus, the HR shown is not after one minute of BMV, which is now stated more clearly in the revised table 2. 

“Mean heart rate during 0-60 seconds after initial recording, median (IQR)»

MINOR COMMENTS

Numbers < 10 should be written in full

This has been corrected. 

Overall, there are too many unnecessary abbreviations in the manuscript that make it difficult to read and digest.

We have replaced FSB and MSB with the phrases “fresh stillbirth” and “macerated stillbirths” throughout. 

Table 1 – data are usually presented as n (%) rather than % (n). As it is not clear what the numbers are for the child characteristics weight ranges it would be best to indicate in the table legend at the bottom of the table that data are presented as % (n) unless indicated otherwise. Any abbreviations (e.g., bpm) need to be explained in the table legend (which goes for other tables also – e.g., MSB in Table 2).

We have changed the tables according to this comment to indicate % (n). The legend includes any abbreviation, and MSB is not used now. 

Table 5 needs to indicate what the numbers are – presume relative risk and 95 % CI?

Correct, Table 5 (now Table 4) is relative risk and 95% confidence interval as now included in the headline.

---

## [Decision Letter · Decision Letter 1]

27 Aug 2020

PONE-D-19-30351R1

Increased perinatal survival and improved ventilation skills over a five-year period: An observational study

PLOS ONE

Dear Dr. Størdal,

Thank you for submitting your manuscript to PLOS ONE. After careful consideration, we feel that it has merit but does not fully meet PLOS ONE’s publication criteria as it currently stands. Therefore, we invite you to submit a revised version of the manuscript that addresses the points raised during the review process.

Please make the minor changes suggested by reviewer #1. Once this is done the paper should be acceptable for publication.

We look forward to receiving your revised manuscript.

Kind regards,

Kelli K Ryckman

Academic Editor

PLOS ONE

Reviewers' comments:

Reviewer's Responses to Questions

**Comments to the Author**

1. If the authors have adequately addressed your comments raised in a previous round of review and you feel that this manuscript is now acceptable for publication, you may indicate that here to bypass the “Comments to the Author” section, enter your conflict of interest statement in the “Confidential to Editor” section, and submit your "Accept" recommendation.

Reviewer #1: All comments have been addressed

Reviewer #2: (No Response)

2. Is the manuscript technically sound, and do the data support the conclusions?

Reviewer #1: Yes

Reviewer #2: Yes

3. Has the statistical analysis been performed appropriately and rigorously? 

Reviewer #1: Yes

Reviewer #2: Yes

4. Have the authors made all data underlying the findings in their manuscript fully available?

Reviewer #1: Yes

Reviewer #2: Yes

5. Is the manuscript presented in an intelligible fashion and written in standard English?

Reviewer #1: Yes

Reviewer #2: Yes

6. Review Comments to the Author

Reviewer #1: Thank you for the opportunity to rereview this paper entitled, “Increased perinatal survival and improved ventilation skills over a five-year period”.

I have reviewed the responses and changes made, and am satisfied with the revised manuscript.

Reviewer #2: Thanks for responding to our prior editorial comments. I still have a few questions as indicated on the attached PDF.

I think you should clarify your description of the amount of BMV the subjects received. In methods, you indicate your analyzed this over the first 5 minutes, but you only present data on the first 60 seconds.

Please clarify how "duration of the first BMV sequence increased" & "BMV was continued for an increasing part of the first 60 seconds", & yet the duration of BMV did not increase?

In Table 5, you refer to a "PEEP-study" but you've eliminated any other mention of this study - you need to clarify that.

There is a truly dramatic increase in the amount of "stimulation" used for resuscitation (Table 2); please comment on that. Was this in part due to the training?

7. PLOS authors have the option to publish the peer review history of their article (what does this mean?). If published, this will include your full peer review and any attached files.

Reviewer #1: No

Reviewer #2: No

---

## [Author Response · Author response to Decision Letter 1]

15 Sep 2020

Please find attached rebuttal letter in a separate file.

---

## [Editor Report · Decision Letter 2]

29 Sep 2020

Increased perinatal survival and improved ventilation skills over a five-year period: An observational study

PONE-D-19-30351R2

Dear Dr. Størdal,

We’re pleased to inform you that your manuscript has been judged scientifically suitable for publication and will be formally accepted for publication once it meets all outstanding technical requirements.

Kind regards,

Kelli K Ryckman

Academic Editor

PLOS ONE
---

## [Editor Report · Acceptance letter]

1 Oct 2020

PONE-D-19-30351R2 

Increased perinatal survival and improved ventilation skills over a five-year period: An observational study 

Dear Dr. Størdal:

I'm pleased to inform you that your manuscript has been deemed suitable for publication in PLOS ONE. Congratulations! Your manuscript is now with our production department. 

Kind regards, 

on behalf of

Dr. Kelli K Ryckman 

Academic Editor

PLOS ONE